# Toxicity and Toxin Composition of the Greater Blue-Ringed Octopus *Hapalochlaena lunulata* from Ishigaki Island, Okinawa Prefecture, Japan

**DOI:** 10.3390/toxins11050245

**Published:** 2019-04-29

**Authors:** Manabu Asakawa, Takuya Matsumoto, Kohei Umezaki, Kyoichiro Kaneko, Ximiao Yu, Gloria Gomez-Delan, Satoshi Tomano, Tamao Noguchi, Susumu Ohtsuka

**Affiliations:** 1Laboratory of Marine Bioresource Chemistry, Graduate School of Biosphere Science, Hiroshima University, Higashi-Hiroshima 739-8528, Japan; m186510@hiroshima-u.ac.jp (K.U.); m181965@hiroshima-u.ac.jp (K.K.); m171023@hiroshima-u.ac.jp (X.Y.); 2Faculty of Human Culture and Science, Prefectural University of Hiroshima, Hiroshima 734-8558, Japan; takuya62@pu-hiroshima.ac.jp; 3Department of Fisheries, Cebu Technological University-Carmen Campus, 6005 Cebu, Philippines; glogdelan@gmail.com; 4Department of Ecology and Evolutionary Biology, University of California, Los Angeles 90095, CA, USA; tomano@ucla.edu; 5Faculty of Healthcare, Tokyo Health Care University, Tokyo 154-8568, Japan; t-noguchi@thcu.ac.jp; 6Takehara Station, Setouchi Field Science Center, Graduate School of Biosphere Science, Hiroshima University, Takehara City, Hiroshima 725-0024, Japan; ohtsuka@hiroshima-u.ac.jp

**Keywords:** greater blue-ringed octopus, *Hapalochlaena lunulata*, posterior salivary gland, paralytic toxicity, Ishigaki Island, tetrodotoxin, LC-MS

## Abstract

The toxicity of the greater blue-ringed octopus *Hapalochlaena lunulata*, whose bite is fatal to humans, was examined to better understand and prevent deaths from accidental bites. Living specimens were collected from tide pools on Ishigaki Island, Okinawa Prefecture, Japan, in November and December of 2015, 2016, and 2017. The specimens were examined for the anatomical distribution of the toxicity, which was expressed in terms of mouse units (MU), by the standard bioassay method for tetrodotoxin (TTX) in Japan. Paralytic toxicity to mice was detected in all of the soft parts. The posterior salivary glands exhibited the highest toxicity score with a maximum level of 9276 MU/g, which was classified as “strongly toxic” (more than 1000 MU/g tissue) according to the classification of toxicity established by the Ministry of Health, Labor and Welfare of Japan, followed by the hepatopancreas (21.1 to 734.3 MU/g), gonads (not detectable to 167.6 MU/g), arms (5.3 to 130.2 MU/g), and other body areas (17.3 to 107.4 MU/g). Next, the toxin from the salivary glands was partially purified by a Sep-Pak C18 cartridge and an Amicon Ultra Centrifugal Filter with a 3000-Da cut-off, and analyzed by liquid chromatography-mass spectrometry (LC-MS) equipped with a φ2.0 × 150-mm (5 μm) TSKgel Amide-80 column (Tosoh, Tokyo, Japan) with a mixture of 16 mM ammonium formate buffer (pH 5.5) and acetonitrile (ratio 3:7, *v*/*v*) as a mobile phase. This study aimed to clarify the toxicity and the composition of TTX and its derivatives in this toxic octopus. The main toxin in this toxic octopus was identified as TTX, along with 4-*epi* TTX, 4, 9-anhydroTTX and 6-*epi* TTX. Further, the toxicity of this species is also significant from a food hygiene point of view.

## 1. Introduction

It has been known for a long time that several species of octopus secrete from the posterior salivary glands a substance that is toxic to prey organisms [1,2]. In humans, the bites of several species of octopuses can cause pain around the wound area [3,4,5], and occasionally even fatal lesions. Symptoms of such lesions include numbness of the face as well as acute and progressive skeletal muscle weakness due to the venom from the posterior salivary glands, which is connected to the beak. The posterior salivary glands are the typical storage sites of octopus venoms. Venomous bites by octopuses belonging to the genus *Hapalochlaena* (Cephalopoda: Octopoda: Octopodidae) are among the most dangerous octopus bites, and there have been several reported fatalities and near fatalities resulting from their bites. In Australia, many case reports on bites by toxic octopuses have been recorded [6,7,8,9,10,11,12]. Fortunately, bites by this species have not been reported in Japan. The potent neurotoxin contained in the posterior salivary glands of *Hapalochlaena maculosa* (formerly *Octopus maculosus*) that is responsible for fatal bites has been identified as tetrodotoxin (TTX), formerly known as maculotoxin [13,14,15,16,17]. Consequently, TTX have become the major point of focus on cephalopod toxinological research. It is well-known that TTX is an extremely potent low-molecular weight neurotoxin (C_11_H_17_N_3_O_8_; MW = 319) that is associated with neurotoxic marine poisonings; it has powerful pharmacological action to block the specific voltage-dependent sodium channels of biological membranes [18]. TTX is one of the most powerful marine biotoxins, and similar to saxitoxin, which is a paralytic shellfish poison, it has a 50% lethal dose (LD_50_) in mice of 10 µg/kg, as compared to 10,000 µg/kg for sodium cyanide [19]. TTX is a heterocyclic guanide compound whose chemical structure has been characterized. The TTX analogs isolated from puffers can be classified into four groups: (1) analogs chemically equivalent to TTX (4-*epi*TTX and 4,9-anhydroTTX); (2) deoxy analogs (5-deoxyTTX, 11-deoxyTTX, etc.); (3) 11-CH_2_OH oxidized analogs (11-oxoTTX); and (4) C11-lacking analogs (11-norTTX-6(*S*)-ol and 11-norTTX-6(*R*)-ol) [20,21]. The pharmacological properties of these TTX analogs were revealed to be closely similar to those of TTX.

Human intoxication with this toxin is characterized by symptoms of respiratory failure that result in death in the most severe cases, e.g., TTX poisoning cases due to ingestion of puffer fish. Since there is no cure for TTX poisoning, the mortality rate is very high. The genus *Hapalochlaena* consists of three toxic species: the blue-lined octopus *H. fasciata*, the lesser blue-ringed octopus *H. maculosa*, and the greater blue-ringed octopus *H. lunulata*, all of which are distributed in the tropical to subtropical zones of the Indo-West Pacific, and are especially common in marine waters around Australia. The geographical distribution of *H. lunulata* extends to Indonesia, the Philippines, Papua New Guinea, Vanuatu, the Solomon Islands, and even Japan [22]. Due to its high level of TTX, the greater blue-ringed octopus *H. lunulata* is regarded as one of the most venomous marine animals in the world. However, the toxicity and toxin composition of the greater blue-ringed octopus *H. lunulata* in Japan remain unknown. There is an urgent need to clarify the toxicity and toxin composition of octopuses belonging to the genus *Hapalochlaena* (Cephalopoda: Octopoda: Octopodidae) in Japan, which prompted us to undertake the present study. From the saliva or salivary glands, many physiologically active substances have been separated as salivary gland toxins in cephalopods [2], but this study aimed to clarify the toxicity and the composition of TTX and its derivatives in the greater blue-ringed octopus from Ishigaki Island, Okinawa Prefecture, Japan, as part of a series of studies on the toxification mechanism of TTX-bearing octopuses.

## 2. Results and Discussion

Table 1 shows the anatomical distribution of the toxicity of the octopus. In the mouse bioassay for lethal potency described below, all seven specimens showed paralytic toxicity, irrespective of the date of collection. The posterior salivary glands exhibited the highest toxicity score with a maximum level of 9276 mouse units (MU)/g (total toxicity per specimen, 234.8 MU), which was classified as “strongly toxic” (more than 1000 MU/g tissue) according to the classification of toxicity established by the Ministry of Health, Labor and Welfare of Japan, followed by the hepatopancreas (21.1 to 734.3 MU/g), gonads (not detectable to 167.6 MU/g), other body parts (17.3 to 107.4 MU/g), and arms (5.3 to 130.2 MU/g). Due to the high levels of TTX, the greater blue-ringed octopus *H. lunulata* is regarded as one of the most venomous marine animals. The toxicity of the whole body was assessed in two specimens; they exhibited a toxicity of 74.3 and 100.4 MU/g, and the range of total toxicity per specimen was 61.7 to 234.8 MU/g. These findings suggest that among all of the body parts of *H. lunulata* in Japan, the posterior salivary glands have the highest toxicity, although the toxicity was not exclusively localized in this organ. In contrast, when the toxicity of the posterior salivary glands and other different body parts were examined in another study using specimens of *H. maculosa* collected in the Philippines [23], the posterior salivary glands exhibited a toxicity of 274 MU/g (total toxicity per specimen, 41 MU) while the other body parts exhibited a toxicity of 11 MU/g (total toxicity per specimen, 133 MU); in other words, the toxin was distributed mainly in the other soft body parts, and not in the posterior salivary glands. In the present study, there were wide inter-specimen variations in the toxicity of the greater blue-ringed octopus *H. lunulata*, and correct identification of this species is difficult. The misidentification of *H. lunulata* is a potential risk to human safety as it may lead to fatal bites and severe morbidity, especially if it is misidentified as an edible octopus. In fact, food poisoning incidents due to the ingestion of TTX-bearing octopus *H. fasciata* have been reported in Taiwan [24].

In this context, in 2016 and 2018, blue-lined octopus *H. fasciata* specimens were collected in Japan, one in Lake Hamana (brackish water lake), Shizuoka Prefecture, and one off the coast of Boso Peninsula, Chiba Prefecture [25]. Although the toxicity of the posterior salivary glands was below the level of detection in one octopus and 42.5 MU/g in the other octopus, the fact that *Hapalochlaena* spp. were found in the coastal waters of Japan proper within the temperate zone may indicate that TTX-bearing octopuses could become a serious health issue in Japan in the near future. Further, the toxicity of these species is also significant from a food hygiene point of view.

In humans, the minimum lethal dose of TTX is estimated to be approximately 10,000 MU (1 MU = 0.178 µg), which is equivalent to 2 mg of TTX crystal [26]. TTX is heat-resistant and does not decompose during general cooking processes, such as heating and boiling, and there are presently no known antidotes or antitoxins to TTX. Therefore, treatments for TTX poisoning are considered to be only supportive. TTX poisoning is characterized by a few symptoms in the victims. The symptoms depend on the amount of toxin ingested as well as the age and health of the victim. In Australia, the fatal envenomation of adult green sea turtles by accidental consumption of seagrass and blue-lined octopuses together has also been reported [27].

Figure 1. shows the results of liquid chromatography-mass spectrometry (LC-MS) analysis. The ion-monitored mass chromatograms show protonated molecular ion peaks (M + H)^+^ at *m*/*z* = 320 and 302, which coincided well with data for TTX standards [20,28,29]. TTX and two TTX analogs (4-*epi*TTX and 6-*epi*TTX) all had the same molecular weight (319 Da). The protonated molecular ion peak (M + H)^+^ of 6-*epi* TTX estimated from the literature [29] was also detected. The main component was identified as TTX along with 4-*epi* TTX, 4,9-anhydroTTX, and 6-*epi*TTX. It could be unambiguously concluded from the symptoms in the mice and the results of LC-MS analysis that the toxins contained in *H. lunulata* collected on Ishigaki Island, Okinawa Prefecture, Japan, comprise a mixture of TTX and TTX derivatives. High-resolution LC-MS/MS analysis is also useful to ensure identification of targets [30]. Multiple reaction monitoring (MRM) mass spectral analysis was not examined this time. As for toxins contained in *Hapalochlaena* sp., existence of peptide and protein neurotoxins are reported [31]. These points were not examined this time.

To the best of our knowledge, this is the first report on the toxicity and toxin composition of *H. lunulata* from a subtropical area of Japan. In another study using post-column fluorescence high-performance liquid chromatography, TTX was present in all of the body parts of *H. maculosa* from South Australia, including high concentrations of TTX in the arms, abdomen, and cephalothorax [32]. In contrast, TTX was found only in the posterior salivary glands, mantle tissue and ink of *H. lunulata* from Bali, Indonesia [33]. Octopuses generally have an ink sac. When they encounter an enemy or are attacked by them, they secrete a lot of ink from the sac to help conceal themselves to escape. While TTX is generally accepted to be a powerful chemical defense against predators [19,34,35], offensive functions have also been suggested for the release of TTX. Octopuses are carnivores, and their salivary glands, especially the posterior salivary glands, produce venom to assist them in capturing various crustacean preys, such as shrimps and crabs. In blue-ringed octopuses, TTX may serve as a hunting tool for paralyzing prey as well as a biological defense tool against predation for eggs and young blue-ringed octopuses of planktonic stage [36]. Thus, TTX-bearing octopuses appear to use TTX for capturing prey as well as for protecting themselves from enemies. On the other hand, TTX levels of adult females, paralarvae, and eggs were investigated to ascertain the relationship between maternal and offspring TTX levels, and to examine TTX-ontology through hatching. It is suggested that embryos or their bacterial symbionts begin independent production of TTX before hatching [36]. In this connection, it has demonstrated that TTX levels in the embryos of puffer fish increase until hatching; emphasizing its endogenous origin [37].

TTX is found in a remarkably wide range of marine and terrestrial animals across disparate taxa; it is found not only in pufferfish, but also in a variety of vertebrates and invertebrates [38,39,40]. It is generally accepted that TTX is accumulated in TTX-bearing animals through the food chain, starting from bacteria as the ultimate producers of the toxin, although the exact biosynthetic and metabolic pathways of TTX remain unknown. However, it is also possible that TTX is not obtained via the food chain and is instead produced by symbiotic or parasitic bacteria that directly accumulate inside of the octopuses. It is not clear at present whether the toxins in our toxic octopus specimens were endogenous or exogenous in origin. Since octopuses are generally carnivorous feeders, it is more plausible that octopus specimens accumulate the toxin by feeding on toxic marine organisms in the sampling areas. It is not uncommon for toxins to be transported and accumulated in food chains, particularly in marine biota, and feeding experiments may be useful for clarifying the origins of the toxins contained in toxic octopuses. Evidence of TTX-producing bacteria isolated from some TTX-bearing organisms, such as puffer fish and xanthid crabs, exists [41,42]. Although the origin of TTX in the venomous octopuses in this study remains unclear, TTX appears to be produced by bacteria in the posterior salivary glands of *H. maculosa* [23]. In our study, wide inter-specimen variations in toxicity were observed even within the same species (Table 1), suggesting that the level of TTX in toxic octopuses is related to some environmental factors, or that it comes from food. Nonetheless, the mechanism may involve factors other than the food chain. Further research to elucidate the associated mechanisms of toxification is now in progress. In addition, investigations on specimen-, location-, and size-dependent variations in the toxicity of *H. lunulata* are also needed, and results in comparison to those of the present study will be published elsewhere at a later date.

## 3. Materials and Methods

### 3.1. Materials

Figure 2 shows the sampling location off the coast of Ohsaki beach in Ishigaki Island, Okinawa Prefecture, Japan. A total of nine specimens of this octopus, which were found swimming in tide pools, were collected with a small type of landing net. In this sampling area, tide pools suitable for the sampling of these species appeared around midnight in the winter seasons throughout 2015 to 2017. Figure 3 shows a representative specimen of the greater blue-ringed octopus *H. lunulata* that was assessed in this study. This octopus was identified by its morphological characteristics with reference to an illustrated catalog of cephalopod species [22]. The total body length was around 10 cm from the mantle apex to the arm tips. The appearance of this live octopus was characterized by numerous small brilliant blue rings, which it flashed as an aposematic warning signal, scattered on its arms and body. Dr. T. Okutani (Prof. Emeritus, Tokyo University of Marine Science and Technology), a taxonomist of cephalopods, kindly confirmed our species identification. The animals were placed in individual 50 mL polypropylene conical tubes filled with fresh seawater, and transported alive by air from Ishigaki Island to the Laboratory of Marine Bioresource Chemistry, Hiroshima University. After the seawater was discarded, the animals were frozen and stored at –20 °C for no longer than 1 month before analysis. Following identification according to the anatomical location of a pair of posterior salivary glands with reference to previous reports [2,31,43,44], the posterior salivary glands were carefully excised.

### 3.2. Mouse Bioassay for Lethal Potency

To examine the anatomical distribution of toxicity, the specimens were dissected into five parts: the posterior salivary glands, gonads, hepatopancreas, arms, and other body parts. Because the principal toxin was suspected to be TTX, the standard bioassay method for TTX [45] was used with slight modifications. Briefly, the tissues were finely cut with scissors, ground with a mortar and pestle, and then transferred into a glass test tube containing 3 mL of 0.1% acetic acid. The mixture was heated in a boiling water bath for 5 min, cooled, then centrifuged at 11,000× *g* for 10 min at 4 °C. One milliliter of the supernatant or its dilution was intraperitoneally injected into male mice of the ddY strain (18 to 20 g in body weight). Lethality was expressed in terms of MU, where 1 MU was defined as the amount of toxin that kills a mouse in 30 min.

### 3.3. Preparation and Identification of Toxins

The remaining extracted liquid for the mouse bioassay was evaporated until dry, and the sample was then dissolved in a small amount of water. Each sample was centrifuged, and the supernatant was applied to a Sep-Pak C18 cartridge (Waters, Milford, MA, USA) equilibrated with water after washing with MeOH. The unbound toxic fraction was concentrated, freeze-dried, dissolved in a small amount of water, and then ultra-filtered in an Amicon Ultra Centrifugal Filter with a 3000-Da cut-off (Merck Millipore, Cork, Ireland) by centrifugation at 5300× *g* for 15 min at 4 °C. The clear filtrate was used as the sample solution and was subjected to analysis by an electrospray ionization-liquid chromatography-mass spectrometry (ESI-LC/MS) system according to the method of Nakagawa et al. [29]. For LC-MS, an LC system (Agilent 1100 series; Agilent Technologies, Palo Alto, CA, USA) with a φ2.0 × 150 mm (5 μm) TSKgel Amide-80 column (Tosoh, Tokyo, Japan) was used with a mixture of 16 mM ammonium formate buffer (pH 5.5) and acetonitrile (ratio 3:7, *v*/*v*) as a mobile phase at a flow rate of 0.2 mL/min at 25 °C. The column was connected to the electrospray interface of an API2000 quadrupole mass spectrometer (Applied Biosystems, Warrington, UK). Six ions at *m*/*z* 272 (5,6,11-trideoxyTTX), 288 (5,11-dideoxyTTX, 6,11-dideoxyTTX), 290 (11-*nor*TTX-6(*S*)-ol), 302 (4,9-anhydroTTX), 304 (5-deoxyTTX, 11-deoxyTTX) and 320 (4-*epi*TTX, 6-*epi*TTX, TTX), corresponding to the [M + H]^+^ of TTX and its analogs, were detected in selected ion-monitoring (SIM) mode. The elution time of the TTX analogs in the partially purified *H. lunulata* toxins was calculated based on the TTX standards with reference to previously reported elution times [20,28,29]. Reference standard samples of TTX were essentially prepared by chromatography on activated charcoal, Bio-Gel P-2, and Bio-Rex 70 (H^+^ form) from ribbon worm *Cephalothrix simula,* as reported previously [38].

## Figures and Tables

**Figure 1 toxins-11-00245-f001:**
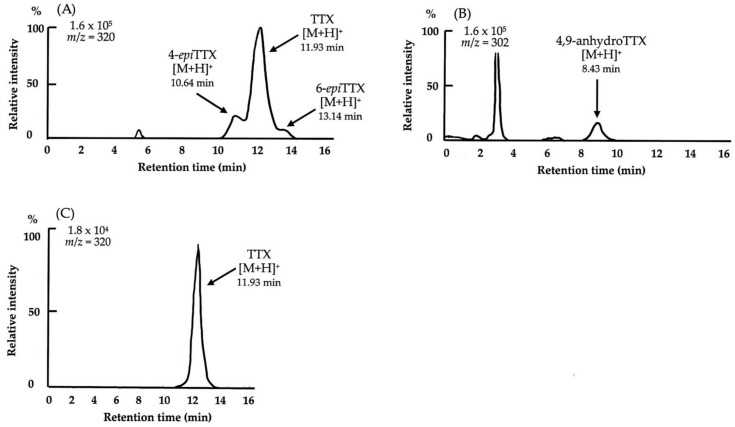
Selected ion-monitored liquid chromatography-mass spectrometry (LC-MS) chromatograms of the toxin from the posterior salivary glands of the greater blue-ringed octopus *Hapalochlaena lunulata*. (**A**,**B**) Toxins from the posterior salivary glands. (**A**) *m*/*z* = 320; (**B**) *m*/*z* = 302; (**C**) Reference standard samples of TTX; *m*/*z* = 320.

**Figure 2 toxins-11-00245-f002:**
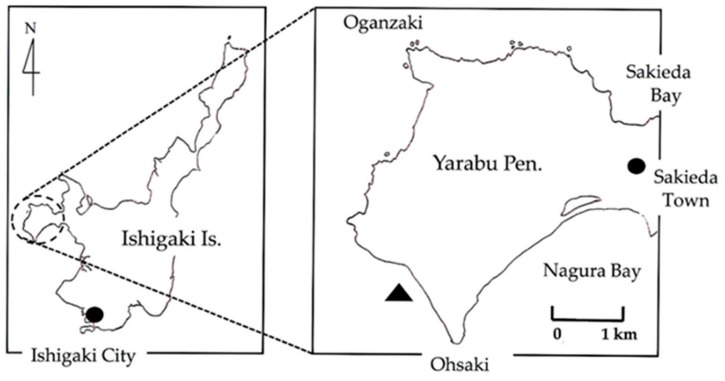
Map showing the location for the collection of greater blue-ringed octopus *Hapalochlaena lunulata* on Ishigaki Island (N24°20′, E124°09′), Okinawa Prefecture, Japan. Ishigaki Island is shown in the map on the left. The map on the right is an enlarged image of Yarabu Peninsula showing the sampling location (▲). ●: Ishigaki City in the map on the left and Sakieda Town in the map on the right.

**Figure 3 toxins-11-00245-f003:**
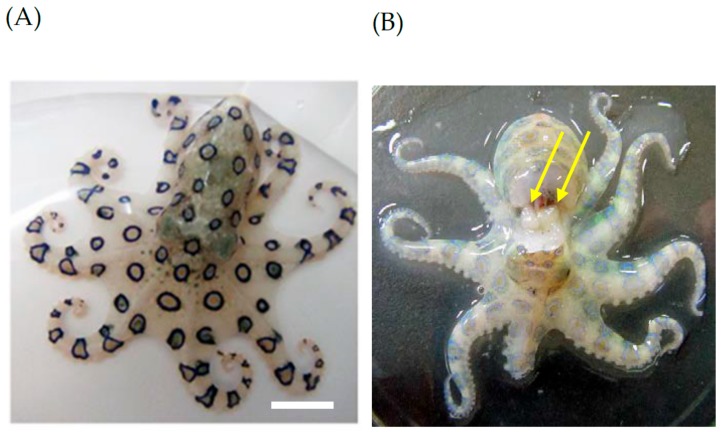
Greater blue-ringed octopus *Hapalochlaena lunulata* from Ishigaki Island, Okinawa Prefecture, Japan. (**A**) Living whole body. White scale bar = 1.0 cm. (**B**) Location of the posterior salivary glands (yellow arrows).

**Table 1 toxins-11-00245-t001:** Anatomical distribution of the toxicity of the greater blue-ringed octopus *Hapalochlaena lunulata* from Ishigaki Island, Okinawa Prefecture, Japan.

**Year**	**2015**					**2016**		
**Date of Collection**	**Nov.28**				**Dec.27**	**Nov.17**		
Weight of whole body (g)	9.63		5.24		0.83	5.30		1.66
Organ	Weight (g)	Toxicity (MU/g)	Weight (g)	Toxicity (MU/g)	Toxicity (MU/g)	Weight (g)	Toxicity (MU/g)	Toxicity (MU/g)
Posterior salivary glands	0.37	288.0	0.02	9276	74.3 *	0.04	704.9	100.4 *
Gonad	0.05	52.3	0.03	ND		0.13	-	
Hepatopancreas	0.38	58.8	0.17	145.4		0.42	21.1	
Arm	3.28	5.3	1.99	9.0		2.85	7.5	
Others	0.60	26.3	0.36	17.3		1.86	68.2	
Total toxicity/sp.	163.9		234.8		61.7	185		167.0
**Year**	**2017**							
**Date of Collection**	**Dec.4**							
Weight of whole body (g)	1.08		3.14		3.65		0.71	
Organ	Weight (g)	Toxicity (MU/g)	Weight (g)	Toxicity (MU/g)	Weight (g)	Toxicity (MU/g)	Weight (g)	Toxicity (MU/g)
Posterior salivary glands	0.02	1729	0.02	1059	0.04	491.1	0.003	5002.9
Gonad	0.02	ND	0.04	167.6	0.06	ND	0.02	ND
Hepatopancreas	0.11	265.4	0.35	195.3	0.15	88.7	0.04	734.3
Arm	0.49	10.7	1.89	21.7	1.07	22.2	0.28	130.2
Others	0.44	21.6	0.84	107.4	2.03	20.7	0.16	80.2
Total toxicity/sp.	78.5		227.0		98.7		93.7	

-: not tested; ND: not detected; *: whole body.

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
