# Peer review of "Toxicity and Toxin Composition of the Greater Blue-Ringed Octopus Hapalochlaena lunulata from Ishigaki Island, Okinawa Prefecture, Japan"

_toxins, 2019, doi:10.3390/toxins11050245_

Reviewer 1 Report

This is a short and rather straightforward piece of work and I see no major problems with it. However, I'd like the authors to address the following four observations:

1.     In addition to TTX, Hapaloclhlaena sp has been reported to also contain peptide and protein neurotoxins, see e.g. J. Mol. Evol. 68, 311-321. This should be mentioned in the context of Hapalochlaena toxicity, although admittedly, TTX is the major problem here.

2.     With regard to point 1 - the authors state that they aim to clarify the toxicity and toxin composition of the greater blue-ringed octopus in this paper, but in fact focus solely measure TTX and derivatives. This should be clarified in the aim of the paper.

3.     It’s good that reference compounds have been used for the LC-MS analysis. However, it has previously been demonstrated in previous work that “false positives” may pop up in TTX analysis (Marine Drugs. 25, 10.3390/md140400063 and a few refs therein). Ideally, an MS/MS spectrum should be taken on these samples. If this is not an option for the authors, uncertainty of TTX MS analysis should be mentioned.

4.     The appearance of the chromatograms is a bit odd – almost looks like the peak lines have been embossed manually. Is this the case? I’d like to be sure that I am looking at the original chromatograms, so I would appreciate if the authors could clarify how they produced the chromatogram pictures in the manuscript.

Apr. 26, 2019

Author Response

Author's Reply to the Review Report (Reviewer 1)

Manuscript ID toxins-486599

Thank you very much for your criticism, advice and instruction.  As grammatical errors, typing mistakes and insufficient explanation of the contents etc. were found as you pointed out, we retyped and revised our manuscript carefully, taking your suggestions into consideration.  Main changes made are as follows.  We hope that all changes we have made will well cover your comments.  Our answers to each suggestions were as follows.

Point 1.

    As for toxins contained in Hapalochlaena sp., marine biotoxins such as peptide and protein neurotoxins are known as you pointed out.  According your suggestion, the following sentence was added to “2. Results and Discussion”.

L128-129.

“As for toxins contained in Hapalochlaena sp., existence of peptide and protein neurotoxins are also reported [31].  These points were not examined this time.”

We cited ref. No.31 as mentioned below.

“31.  Fry, B.G., Roelants, K., Norman, J.A., Tentacles of venom: toxic protein convergence in the kingdom Animalia.  J. Mol. Evol.  2009, 68, 311-321.”

Point 2.

According to your suggestions, aims of this study were added to “Abstract” and “Introduction” in this manuscript.

L21-22.

Changed to, “This study aimed to clarify the toxicity and the composition of TTX and its derivatives in this toxic octopus”.

L67-71

Changed to, “From the saliva or salivary glands, many physiologically active substance have been separated as salivary gland toxins in cephalopods [2], but this study aimed to clarify the toxicity and the composition of TTX and its derivatives in -----.”

Point 3.

   According to your suggestion, the following sentence was added to “2. Results and Discussion”.

L126-128

“High-resolution LC-MS/MS analysis is also useful in order to ensure identification of targets [30].  Multiple reaction monitoring (MRM) mass spectral analysis was not examined this time."

We cited ref. No.30 as mentioned below.

“30. Bane, V., Hutchinson, S., Sheehan, A., Brosnan, B., Barnes, P., Lehane, M., Furey, A., LC-MS/MS method for the determination of tetrodotoxin (TTX) on a triple quadruple mass spectrometer.  Food addit. Contam.: Part A.  2016, 33, 1728-1740.”

Point 4.

Figure 1.

Here, I would like to show you the original chromatograms and figure1 based on them.  As you can see, solid lines in them were not so clear.  Figure 1 based on the original chromatograms was revised faithfully as possible as we can.

Other small grammatical errors were corrected , as suggested.

Reviewer 2 Report

The manuscript entitled “Toxicity and Toxin Composition of the Greater Blue-2 Ringed Octopus Hapalochlaena lunulata from 3 Ishigaki Island, Okinawa Prefecture, Japan” reports, for the first time, toxicity and toxin profile of H. lunulatein Japan coasts.

The study is of high interest and impact as it is the first time that the toxicity profile of H. lunulatein Japan coasts is showed. However, according this reviewer, there are several issues that should be addressed before considering the manuscript for publication in Toxins.

Major points:

1.    Abstract contains details only appropriate for materials and methods section (such as filters). Please, revise it.

2.  Figure 1. Selected ion-monitored LC-MS chromatograms of the toxin from the posterior salivary glands of the greater blue-ringed octopus Hapalochlaena lunulata. (A, B) Toxins from the posterior salivary glands. (A) m/z = 320; (B) m/z = 302; (C) authentic TTX; m/z = 320.  What is the meaning of “authentic”? Standard?

3.    Figure 3. Greater blue-ringed octopus Hapalochlaena lunulata from Ishigaki Island, Okinawa Prefecture, Japan. (A) Live whole body. White scale bar = 1.0 cm. (B) Location of the posterior salivary glands (yellow arrows). Figure 3A refers to a “living” octopus? Is it not dissected?

4.    Authors state that there are no studies explaining the ontogenesis of the TTX in H. lunulateand that the origin of the toxin should be studied in terms of intake by feeding or preexistence. However, Williams et al.already demonstrated the maternal investment and apparent independent production in offspring of H. lunulate, which clearly points to the preexistence of symbiotic microorganisms from the initial phases of the development(J Chem Ecol. 2011 Jan;37(1):10-7. doi: 10.1007/s10886-010-9901-4). This study is referred by the authors, but not regarding the toxin ontology. Please, reformulate discussion taking this into account.

5.    Standard tetrodotoxin was obtained from Cephalothrix simulaas reported in Toxicon 2000, 38, 763-773. However, in that report diverse analogues of TTX were found. Purification method for TTX and purity level obtained should be stated at least to be considered a standard.

Minor points:

1.    English should be revised

2.    Please, revise the use of “live” organism when referring to “living“ organism.

Author Response

Author's Reply to the Review Report (Reviewer 2)

Manuscript ID toxins-486599

Thank you very much for your criticism, advice and instruction.  As grammatical errors, typing mistakes and insufficient explanation of the contents etc. were recognized as you pointed out, we retyped and revised our manuscript carefully, taking your suggestions into consideration. We hope that all changes we have made will well cover your comments.  Our answers to each suggestions were as follows.

Point 1.

Abstract

According to your suggestion, separation conditions in LC were added to “Abstract” in manuscript as mentined below.

L19-21.

Changed to “-----  equipped with a φ2.0 × 150-mm (5 μm) TSKgel Amide-80 column (Tosoh, Tokyo, Japan) was used with a mixture of 16 mM ammonium formate buffer (pH 5.5) and acetonitrile (ratio 3:7, v/v) as a mobile phase.”

Point 2.

Figure 1.

The expression of “authentic TTX” in Fig.1 was changed into “Reference standard samples of TTX” according to your suggestion.

Point 3.

Figure 3.

Figure 3A is a “living octopus”.  After sampling of this octopus, we took a picture of this living specimen in a paper bowl filled with seawater.

Point 4.

     According to your suggestions, the following sentences were added.  We added one more related ref. No.37 as mentioned below.

“37. Matsumura, K., Production of tetrodotoxin in puffer fish embryos.  Environ. Toxicol. Pharmacol. 1998, 6, 217-219.”

L143-148.

“On the other hand, TTX levels of adult females, paralarvae, and eggs to ascertain the relationship between maternal and offspring TTX levels, and to examine TTX-ontology through hatching were investigated.  It is suggested that embryos or their bacterial symbionts begin independent production of TTX before hatching [36].  In this connection, it has demonstrated that TTX levels in the embryos of puffer fish increase until hatching; emphasizing its endogenous origin [37].”

Point 5.

According to your suggestions, some details of purification methods and purification level for TTX from the ribbon worm C.simula. were added.

L229-231.

Changed to “Reference standard samples of TTX were essentially prepared by chromatography on activated charcoal, Bio-Gel P-2 and Bio-Rex 70(H+ from) from ribbon worm Cephalothrix simula, as reported previously [38].”

Other small grammatical errors were corrected, as suggested.

Apr. 26, 2019